# Genetic Diversity for Effective Resistance in Wheat Landraces from Ethiopia and Eritrea to Fungal Diseases and Toxic Aluminum Ions

**DOI:** 10.3390/plants13081166

**Published:** 2024-04-22

**Authors:** Evgeny V. Zuev, Tatiana V. Lebedeva, Olga V. Yakovleva, Maria A. Kolesova, Alla N. Brykova, Natalia S. Lysenko, Lev G. Tyryshkin

**Affiliations:** Federal Research Center N.I. Vavilov All-Russian Institute of Plant Genetic Resources (VIR), Bolshaya Morskaya Str. 42-44, 190000 Saint Petersburg, Russia; e.zuev@vir.nw.ru (E.V.Z.); riginbv@mail.ru (T.V.L.); oly.yakovleva@mail.ru (O.V.Y.); markolesova@yandex.ru (M.A.K.); a.brykova@vir.nw.ru (A.N.B.); n-lysenko@yandex.ru (N.S.L.)

**Keywords:** *Triticum* species, *T. aestivum*, *T. dicoccum*, *T. aethiopicum*, *T. polonicum*, SNB, *Pt*, HLB, *Bgt*, *Pst*, aluminum ions, resistance, tolerance

## Abstract

To reveal genetic diversity for effective resistance to five foliar diseases and toxic aluminum ions, the entire collection of wheat species from the N.I. Vavilov All-Russian Institute of Plant Genetic Resources (VIR) originating from Ethiopia and Eritrea were studied regarding their traits. The collection contains 509 samples of four wheat species (*Triticum aestivum*—122 samples; *T. aethiopicum*—340 samples; *T. polonicum*—6 samples; and *T. dicoccum*—41 samples). The majority of accessions are new entries of landraces added to the Vavilov collection as a result of the Russian–Ethiopian expedition in 2012. Wheat seedlings were inoculated with causal agents of leaf rust (*Pt*), powdery mildew (*Bgt*), Septoria nodorum blotch (SNB), and dark-brown leaf spot blotch (HLB). The types of reaction and disease development were assessed to describe the levels of resistance. All samples of *T. aethiopicum* were also screened for seedling and adult resistance to *Pt*, *Bgt*, and yellow rust (*Pst*) under field conditions after double inoculation with the corresponding pathogens. To study tolerance to abiotic stress, seedlings were grown in a solution of Al^3+^ (185 µM, pH 4,0) and in water. The index of root length was used to characterize tolerance. Seedlings belonging to only two accessions out of those studied—k-68236 of *T. aethiopicum* and k-67397 of *T. dicoccum*—were resistant to *Pt* at 20 °C but susceptible at 25 °C. Specific molecular markers closely linked to the five genes for *Pt* resistance effective against populations of the pathogen from the northwestern region of Russia were not amplified in these two entries after PCR with corresponding primers. Four entries of *T. dicoccum*—k-18971, k-18975, k-19577, and k-67398—were highly resistant to *Bgt*. All samples under study were susceptible to HLB and SNB. Under field conditions, 15% of the *T. aethiopicum* samples were resistant to *Pst*, both at the seedling and the flag leaf stages, but all were susceptible to the other diseases under study. Among the evaluated samples, 20 entries of *T. aestivum*, 1 of *T. polonicum* (k-43765), and 2 of *T. dicoccum* (k-18971, k-67397) were tolerant to aluminum ions. The identified entries could be valuable sources for the breeding of *T. aestivum* and other wheats for resistance to biotic and abiotic stresses.

## 1. Introduction

Biotic and abiotic stresses are important factors that reduce yield and its quality in wheat (*Triticum* L.). Leaf rust (*Pt*) (*Puccinia triticina* Erikss.), Septoria nodorum blotch (SNB) (*Parastagonospora nodorum* (Berk.) Quaedvlieg, Verkley, and Crous syn. *Stagonospora nodorum* (Berk.) Castell. et Germano), dark-brown leaf spot blotch, also known as Helminthosporium leaf blotch (HLB) (*Bipolaris sorokiniana* (Sacc.) Shoemaker, teleomorph *Cochliobolus sativus* (Ito et Curib.), powdery mildew (*Bgt*) (*Blumeria graminis* (DC.) E.O. Speer f. sp. *tritici* Em. Marchal syn. *Erysiphe graminis* f. sp. *tritici*), and stripe rust (*Pst*) (*Puccinia striiformis* Westend. f. sp. *tritici* Erikss.) are devastating diseases affecting cultivated wheat in many regions, with their widespread development resulting in significant yield losses. The grain yield losses resulting from severe *Pt* epidemics are estimated to be more than 50% if fungicides are not applied [1]. The losses due to SNB can be up to 50% [2]. Susceptible varieties under favorable conditions for HLB can lose up to 100% of their yield [3]. The yield losses from *Bgt* under severe epiphytotic conditions can reach 50% or more [4]. Under favorable conditions in a single field, yellow rust can lead to 100% crop losses [5]. In addition to yield losses, disease development reduces the quality of wheat [2,6,7,8,9]. Acidic soils with aluminum ions predominate in many regions, and the cultivation of wheat in these areas results in a disruption of metabolic processes, the formation of generative organs, a decrease in biomass [10,11,12], and, consequently, a reduction in yield [12,13,14]. In cereals, aluminum toxicity causes a yield loss of at least 30–40% [15].

The cultivation of resistant/tolerant varieties is well known to be an effective and environmentally safe method to reduce losses in wheat production due to biotic and abiotic stresses. To breed such varieties, effective genes and genetic systems for resistance and tolerance are needed.

The collection of the N.I. Vavilov All-Russian Institute of Plant Genetic Resources (VIR, Saint Petersburg, Russia), originally founded by N.I. Vavilov, possesses good prospective samples to identify sources of valuable characteristics for breeding [16,17]. However, previous studies with bread wheat and its relatives showed very narrow genetic diversity for resistance to modern populations of *Pt*, HLB, SNB, and *Bgt* from Russia [18,19,20,21].

Effective resistance to *Pt* in samples from the collection is only controlled by the genes *Lr9*, *Lr19*, *Lr24*, *Lr41*, and *Lr47* [18,19], but the first three of these have long since lost their effectiveness in many parts of the Russian Federation [22,23,24,25]; virulence against *Lr41* and *Lr47* is now described in Russia with significant frequencies [26,27,28]. No bread wheat accession from the VIR collection exhibited high resistance to SNB and HLB [18,19]. The number of effective *Bgt* genes in the Russian Federation is extremely limited [18,29,30]. Only four genes for yellow rust resistance (*Yr5*, *Yr10*, *Yr15*, and *Yr24)* were effective against *Pst* in the Russian Federation [31], but now virulence against the gene *Yr24* has been demonstrated in the south of Russia [32].

The tolerance of bread wheat from the VIR collection to Al^3+^ was studied for more than a thousand accessions, and samples with a high level of the trait were found, but with only a low frequency [33,34,35]. So, the search for new samples with a high tolerance to this abiotic stressor is of great interest. The entries from Ethiopia and Eritrea could be promising material because the soils in these countries have a stable acidic state throughout the entire profile [36,37].

The very low diversity for a high level of resistance/tolerance in cultivated wheats demonstrates that its broadening is an important task. To achieve this task, the evaluation of new plant material from the collection should be performed. Samples of different wheat species from Ethiopia and Eritrea could be of special interest.
This region is the secondary gene center of wheat’s origin [38], and so we could expect an increased frequency of samples with a high level of resistance to diseases [39].In many areas of these countries, acid soils predominate [36], and the selection of tolerant plants could take place during the durable growing of landraces.Three main VIR expeditions were performed to sample wheat landraces from Ethiopia and Eritrea: the first one was performed by N.I. Vavilov in 1927; the second was performed by F.F. Sidorov in 1959; and the third was conducted in 2012. The participants of this third expedition were A.M. Kudryavtsev, Yu.A. Stolpovsky (Institute of General Genetics, Moscow), N.P. Goncharov (Institute of Cytology and Genetics, Novosibirsk) and E.V. Zuev (VIR, St. Petersburg). These new entries have never been studied for the abovementioned traits.

Currently, in the VIR collection, 62 accessions of spring bread wheat from Ethiopia are stored. Most landraces were collected during VIR expeditions, namely those of N.I. Vavilov (1927), F.F. Sidorov (1959), and the mission completed in 2012. Also, in 1980, ten samples were included in the collection from the Ethiopian National phytopathological laboratory (t. Ambo, Ethiopia). Seven landraces were received from the Institute of Biosciences and Bioresources (IBBR-CNR, Bari, Italy) (FAO expedition, 1973). Spring bread wheat from Eritrea is represented by 60 accessions. Vavilov collected eight landraces, while Sidorov collected six. The rest of the entries came from the Nordic Gene Bank in 2019.

In the VIR, traditionally, *T. aethiopicum* Jakubz., *T. dicoccum* (Schrank.) Schuebl., and *T. polonicum* L. are regarded as separate species [40,41]. In another classification by J. Mac Key [42], they are subspecies of *T. turgidum* L.

The VIR collection possesses 326 samples of *T. aethiopicum* from Ethiopia (ETH) and 14 samples from Eritrea (ERI). Most of the samples of this species were collected in the territory of Ethiopia during VIR expeditions: N.I. Vavilov (1927) collected 96 samples, F.F. Sidorov (1959) collected 40 samples, and the 2012 mission collected 110 samples. Eighty samples were obtained from the Institute of Biosciences and Bioresources. Eleven samples of the species from Eritrea were collected by a VIR expedition in 1959.

*T. dicoccum* from Ethiopia was generally collected by N.I. Vavilov in 1927 (23 samples), one sample came into the collection after the F.F. Sidorov expedition, and four samples were collected in 2012. Polish wheat (*T. polonicum*) was collected by the Vavilov expedition (one sample) and by Sidorov (one sample). Other samples came from Germany (Leibniz Institute of Plant Genetics and Crop Plant Research, IPK, Gatersleben).

As mentioned above, broadening the genetic diversity of cultivated wheats for resistance to biotic and abiotic stresses is a very important task, and *Triticum* L. species growing in Ethiopia and Eritrea have great importance in the achievement of this task. So, the general aim of this study was to evaluate the reaction to five diseases and the acid tolerance of *Triticum* species from Ethiopia and Eritrea stored in the VIR collection and to select samples with a high level of each trait’s expression.

## 2. Materials and Methods

### 2.1. Plant Material

Five hundred and nine samples of *Triticum* species (three tetraploids and one hexaploid) from the VIR collection (Table 1) were included in this study. We used the classification of the *Triticum* genus adopted by the VIR [43,44].

The sampling locations for different wheat species in Ethiopia and Eritrea are presented in Figure 1, Figure 2 and Figure 3. To create the maps, ArcView GIS31 was used.

Seedling resistance to leaf rust, Septoria nodorum blotch, powdery mildew, and dark-brown leaf spot blotch was evaluated in all sets of samples. Adult resistance to the diseases was evaluated only in samples of *T. aethiopicum*. Aluminum tolerance was studied for *T. aestivum*, *T. polonicum*, and *T. dicoccum*.

### 2.2. Pathogen Material

Phytopathogens were isolated from the leaves of several susceptible wheat varieties grown in Pushkin Experimental Field (Russia, Northwest Region) in 2022–2023.

Under laboratory conditions, *Pt* populations were maintained and multiplied on seedlings of susceptible cv. Leningradka (k-47882) in a light chamber (2500 lux; temperature—20–22 °C; 24 h of daylight). The characteristics of the populations regarding virulence against 50 *Pt* resistance genes were reported previously [20,21]; only 5 genes (*Lr9*, *Lr19*, *Lr24*, *Lr39* (=*Lr41*), and *Lr47*) were effective against these populations.

*B. sorokiniana* and *S. nodorum* isolation, multiplication, and preparation for plant inoculation were performed as reported in [20,21]. 

The populations of *Bgt* were maintained in a greenhouse at 22–25 °C on intact seedlings of spring wheat cv. Diamant (k-25019). The population characteristics for virulence against 23 *Pm* genes were presented previously [21].

The populations of *Pst* under field conditions were virulent/avirulent against near-isogenic lines of wheat cv. Avocet and wheat samples with the resistance genes *Yr1*, *Yr3c*, *Yr6*, *Yr7*, *Yr9*, *Yr10*, *Yr14*, *Yr15*, *Yr18*, *Yr20*, *Yr21*, *Yr24*, *Yr25*, *Yr26*, *Yr27*, *Yr29*, *Yr30*, *Yr31*, *Yr32*, and *YrSp*/*Yr5.*

### 2.3. Screening of Seedling Resistance to Diseases

Seedling resistance to *Pt*, HLB, SNB and *Bgt* was studied under laboratory conditions as reported in [20]; in each experiment, 15–25 plants were evaluated (the concentration of *P. triticina* spores was 3 × 10^4^ mL^−1^, that of *B. sorokiniana* conidia was 3 × 10^4^ mL^−1^ and that of *S. nodorum* spores was 5 × 10^6^ mL^−1^).

The types of reaction to *P. triticina* and *B. graminis* infections were scored on the 12th day after inoculation according to generally accepted scales [43,44]; disease ratings of HLB and SNB were scored on the 7th day after inoculation on a scale reported in [22,23].

Accessions identified as possessing resistance to a certain disease after the first evaluation were screened for the traits in at least 3 additional independent experiments. Cv. Leningradka was a susceptible control and was sown after each of the 5 experimental samples. Samples of wheat evaluated as resistant to leaf rust at 20 °C were assessed for resistance at 25 °C.

To study juvenile resistance to *Pst*, seedlings of *T. aethiopicum* under field conditions (late sowing) were sprayed with a pathogen spore water suspension. The types of reactions to *Pst* were scored on the 12–14th day after inoculation according to a scale [45] with some modifications.

### 2.4. Identification of Effective Lr Genes with Molecular Markers

Molecular markers of effective genes for *Pt* resistance (Table 2) were used to find these genes in resistant samples of *T. aethiopicum* and *T. dicoccum.* All these genes were transferred to the bread wheat genome from related species (*Aegilops umbellulata* Zhuk., *Thinopirum ponticum* (Podp.) Barkworth & D.R. Dewey, *Th. ponticum*, *Ae. tauschii* Coss., and *Ae. speltoides* Tausch, respectively) [46].

DNA was isolated from seedlings according to [47,48]. The nucleotide sequences of primers are listed in Table 2. The polymerase chain reaction (PCR) was performed according to the original protocols [49,50,51,52,53]. Amplicons were separated on an agarose gel, stained in a solution of ethidium bromide (0.5 mg L^−1^) and visualized under UV light. GeneRuler™ 100 bp Plus DNA Ladder (Fermentas, Waltham, MA, USA) was used to estimate the size of PCR amplified fragments. Th*Lr9*, Th*Lr19*, Th*Lr24*, KS90WGRC10 (*Lr41*), and Pavon *Lr47* were used as controls.

**Table 2 plants-13-01166-t002:** Primer sequences used for PCR with wheat samples’ DNA.

Gene, Chromosome Localization	Marker	Primer Sequence (5′-3′)	Fragment Size (bp)	Reference
*Lr9-*6B	SCS5_550_	TGC GCC CTT CAA AGG AAG TGC GCC CTT CTG AAC TGT AT	550	[49]
*Lr19-*7D	Gb	CAT CCT TGG GGA CCT C CCA GCT CGC ATA CAT CCA	130	[50]
*Lr24-*3D	SCS1302_607_	CGC AGG TTC CAA ATA CTT TTC CGC AGG TTC TAC CTA ATG CAA	607	[51]
*Lr41-*1D	GDM35	CCT GCT CTG CCC TAG ATA CG ATG TGA ATG TGA TGC ATG CA	190	[52]
*Lr47-*7A	PS10	GCT GAT GAC CCT GAC CGG T TCT TCA TGC CCG GTC GGG T	282	[53]

### 2.5. Screening of Seedling Resistance to Toxic Aluminum Ions

The resistance of wheat samples to toxic aluminum ions was studied at the early stages of plant development using the root test method with the calculation of the index of root length (IRL) [54,55]. Twenty to thirty seedlings of the studied samples were grown in special germinator boxes (Figure 4) in an aqueous solution of AlCl_3_ × 6H_2_O (Al^3+^ concentration 185 μM, pH 4.0) and in distilled water with pH 6.5 (control). Each sample was studied with three replicates.

Each germinating box additionally contained resistant tester varieties of bread wheat cv. Atlas 66 (k-44947, IRL 0.66) and cv. Leningradka (IRL 0.65).

The lengths of the roots of the seventh day seedlings were measured, and IRL was calculated as the quotient of the mean length of the longest roots in the experiment and in the control. The studied wheat samples were classified into five resistance groups [55]: 1—ID > 0.81, highly resistant; 2—ID 0.61−0.80, resistant; 3—ID 0.41−0.60, moderately resistant; 4—ID 0.31−0.40, moderately sensitive; and 5—ID < 0.30, susceptible ones. 

### 2.6. Adult Plant Resistance Screening

The adult plant resistance of *T. aethiopicum* to five diseases was studied in 2022 and 2023 in the Pushkin Experimental Field. Twenty to thirty plants from each sample were assessed for the disease’s development.

To study adult resistance to *Pst*, *Pt*, and *Bgt*, samples were sprayed with pathogen spore water suspensions. The types of reaction to *P. triticina*, *P. striiformis* f. sp. *tritici*, and *B. graminis* f. sp. *tritici* were scored on the 12–14th day after inoculation according to generally accepted scales [43,44,45] with modifications. SNB and HLB resistance was screened using the microchamber method of infection [20]. Disease ratings were scored according to the above-presented scale [20].

## 3. Results

### 3.1. Juvenile Resistance in Samples of Wheat Species to the Diseases

Most samples under study, as per the results of three to four independent experiments, were susceptible to all diseases under study at the seedling stage.

Only two Ethiopian accessions out of those studied—k-68236 of *T. aethiopicum* collected in the Yismala region (Amhara), E 36 55, N 11 35 and k-67397 of *T. dicoccum* (v. Adal (Harari), E 42 16, N 9 33)—showed resistance to *Pt* at 20 °C. The disease development on these entries is depicted in Figure 5.

Wheat samples with the genes *Lr9*, *Lr19*, *Lr24*, *Lr41*, and *Lr47* were resistant to the *Pt* populations used in this study (see Section 2. Material and Methods). The STS, microsatellite, SCAR, and CAPS markers, linked to the genes [49,50,51,52,53], were used to identify these genes in resistant samples of wheat species from ETH and ERI. After PCR with specific primers for molecular markers closely linked to these genes, amplicons were visualized only in control samples, but not in k-68236 and k-67397. The electropherograms of the amplification products are presented in Figure 6, Figure 7, Figure 8, Figure 9 and Figure 10.

Both samples, k-68236 of *T. aethiopicum* and k-67397 of *T. dicoccum*, were susceptible to *Pt* at 25 °C (Figure 5).

A high level of resistance to *Bgt* (types of reaction 0-0;) was identified only in entries of *T. dicoccum* from Ethiopia: k-18971 was collected in t. Harer, E 42 07, N 09 18; k-18975 was collected in v. Gara Muleta (Harari), E 41 43, N 09 05, k-19577 (Oromia, 52 km to NW from t. Fiche. h = 2536. E 38 18, N 10 01); and k-67398 was collected in Gamo Gofa region (Southern Nations, Nationalities, and Peoples), E 37 56, N 06 25. Powdery mildew development on the seedlings of these samples is shown in Figure 11.

All samples of the four wheat species under study were highly susceptible to HLB and SNB at the seedling stage, including those of *T. dicoccum* and *T. aethiopicum* previously described as resistant to HLB [56] The diseases’ development on wheat seedlings is presented in Figure 12 and Figure 13.

Fifty-five samples of *T. aethiopicum* were not affected by yellow rust at the seedling stage under field conditions for two subsequent years. These samples and their collection sites are shown in Table 3. *Pst* development on seedlings of three samples is shown in Figure 14.

### 3.2. Adult Resistance in Samples of T. aethiopicum to the Diseases

All samples of *T. aethiopicum* were highly susceptible at the flag leaf stage to *Pt*, *Bgt*, SNB, and HLB. All entries classified as resistant to *Pst* at the seedling stage were not affected by *P. striiformis* f. sp*. tritici* at the flag leaf stage (Table 3).

### 3.3. Toxic Aluminum Ion Tolerance in Wheats

Among the 122 entries of spring bread wheat from Eritrea and Ethiopia, samples with different levels of resistance to toxic aluminum ions were identified. A wide range of hereditary variability for this trait was revealed (IRL = 0.47–0.98). The distribution of samples into the resistance groups is presented in Figure 15—forms belonging to the fourth and fifth groups were not identified.

Twenty samples of *T. aestivum* were classified as highly resistant (Table 4), and the other samples belonged to the second (resistant) and third (moderately resistant) groups (Figure 15). The differences in root lengths in highly resistant and moderately resistant samples after the test for aluminum tolerance are shown in Figure 16.

The range of variability for IRL in forty-one *T. dicoccum* samples from Ethiopia was 0.31–0.91. The studied samples belonged to five resistance groups (Figure 17). Only two highly resistant samples to toxic aluminum ions were found—k-67397 (IRL = 0.91, v. Adal (Harari), E 42 16, N 9 33) and k-18971 (IRL = 0.88, collected in t. Harer, E 42 07, N 09 18).

Among the six studied entries of *T. polonicum* from Ethiopia, only one—k-43765, Oromia, near Addis Ababa. E 38 44 N 09 01—was classified as resistant to Al^3+^ (ILR = 0.72), while others were moderately resistant.

## 4. Discussion

After describing the peculiarities of Ethiopian wheats after his expedition in 1927, N.I. Vavilov [57] noted they were generally tetraploid wheats. Bread wheat was found only as an admixture. After the expedition, Vavilov named tetraploid naked wheat as a separate species, *T. abyssinicum* Vav. Later, M.M. Yakubtsiner renamed it *T. aethiopicum* Jakubz. The characteristic trait of this species is the presence of purple grain genotypes. Up to now, world gene banks have faced problems with its identification—sometimes it is referred to as durum wheat, and in other cases it is referred to as bread wheat.

The results of the 2012 expedition showed that in almost all the points observed by Vavilov, local tetraploid wheats were found mainly as an admixture with breeding varieties, but sometimes as pure crops. *T. aethiopicum* landrace fields appear at an altitude of 2400–2600 m above sea level. Local tetraploid wheats are mostly grown in farmers’ fields away from main roads; these fields can only be reached by foot. In the crops of breeding wheat, there are admixtures of landraces of bread and durum wheats. The greatest diversity of *T. aethiopicum* was revealed around the towns of Gondor, Ankober, and Debre Tysge. Near the towns of Ankobera and Gara Mulleta, pure crops of *T. dicoccum* existed. The 2012 expedition did not find any samples of Polish wheat (*T. polonicum*) in Ethiopia.

Foliar diseases (*Pt*, SNB, HLB, *Bgt*, and *Pst*) and toxic aluminum ions significantly reduce wheat yield (up to 100%) and grain quality. The best method to avoid these losses is to develop and grow resistant and tolerant varieties. To create such varieties, breeders must have valuable initial material possessing new effective genes for resistance.

Unfortunately, the genetic diversity of bread wheat for effective resistance to harmful fungal diseases is extremely low. Only five genes have been found to provide a high level of resistance to leaf rust in the Russian Federation [18,19], and they have now been reported to have lost their effectiveness in some regions of Russia [22,23,24,25,26,27,28]. Only genes for resistance to *Bgt Pm12*, *PmSp*, and *PmKu* were described as effective against the pathogen populations from the northwestern part of the Russian Federation [29,30,58], but samples with these genes were found to be susceptible in 2022 [21]. Several varieties of bread wheat were evaluated as resistant to HLB [56,59] and SNB [60,61], but their re-evaluation showed their susceptibility (not published). Only four genes for yellow rust resistance (*Yr5*, *Yr10*, *Yr15*, and *Yr24*) have been described as effective against *Pst* in Russia [31].

The same situation exists with the resistance to abiotic stress under study: more than a thousand accessions and samples were screened, and genotypes with a high level of Al^3+^ tolerance were selected only with a low frequency [33,34,35].

So, the search for new sources and donors of resistance to diseases and toxic aluminum ions is of great interest for wheat breeders and geneticists.

The landraces from Ethiopia and Eritrea in the VIR collection could be promising material for this search for 3 reasons. 1. According to Vavilov, genetic diversity for breeding valuable traits is concentrated in gene centers of crops’ origin [16,17]. The region of Ethiopia and Eritrea is the secondary gene center of *Triticum*’s origin [38,57], and so we could expect an increased frequency of samples with a high level of resistance to diseases [39]. 2. The soils of these countries have a stable acidic state throughout the profile [36,37], and the natural selection of plants tolerant to aluminum toxicity could take place during the durable growth of landraces. 3. The majority of wheat samples from Ethiopia and Eritrea in the VIR collection have never been studied for the abovementioned traits, while bread wheat and its relatives of different origins from the same collection have been studied extensively [18,19,20,21,62,63,64].

Among the 122 landraces of bread wheat under study, all were susceptible to *Pt*, SNB, HLB, and *Bgt*. Earlier, based on the screening of bread wheat landraces of different origins, it was supposed that local varieties of bread wheat are of no interest for breeding for disease resistance [65], and our present data confirmed this supposition. In contrast, in these landraces, we revealed a very high frequency of Al^3+^-resistant genotypes (16.4%). Moreover, no sample belonged to the susceptible or moderately susceptible classes. All bread wheat samples which were highly resistant to Al^3+^ were collected in the highlands, with altitudes ranging from 1755 to 2337 m. The regional state of Asmara, at altitudes above 2000 m, has the largest number of resistant entries. The most resistant was the sample k-68118 (IRL 0.95), found at the highest collection point (2337 m). Earlier, 595 accessions of *T. turgidum* ssp *durum* (=*T. durum*) from Ethiopia were screened for Al^3+^ tolerance using two methods. A total of 21 tolerant, 180 intermediate, and 394 sensitive accessions were identified, but it was then revealed that all samples that were rated as aluminum-tolerant or as having intermediate tolerance were representatives of bread wheat (*T*. *aestivum*) that had contaminated the durum grain stocks [36]. Notably, the concentration of Al^3+^ (10 µM) was much less than in the present work (185 µM). As acidic soils with high Al^3+^ concentrations dominate in many wheat-growing regions, the selected landraces could be of great value for breeding programs aimed at obtaining aluminum-resistant varieties.

Many years ago, the *T. aethiopicum* species was described by Vavilov [57] as being susceptible to leaf rust but including resistant genotypes. This work did not support this conclusion. We identified only one accession of the species (k-68236 from Ethiopia) resistant to leaf rust at the seedling stage at 20 °C. In the scientific literature, the identification of disease resistance genes using molecular markers is often described in samples of species that are not initial donors of the genes [66,67,68,69,70]. This could probably be explained by the presence of orthologous genes in different genomes. After the PCR of DNA from this landrace with primers specific to genes effective in bread wheat, we did not observe any amplicons (Figure 6, Figure 7, Figure 8, Figure 9 and Figure 10). The absence of effective *Lr* genes in the sample was confirmed by its susceptibility both in the seedling stage at 25 °C and under field conditions. All samples of the *T. aethiopicum* species were highly susceptible both in the seedling and in the flag leaf stages to *Bgt*, SNB, and HLB, including entries previously described as being highly resistant to HLB [56]. The possible explanations for these discrepancies are the inoculation of intact seedlings in our study with the HLB causal agent vs. the inoculation of leaf segments on benzimidazole in the cited work, and different timeframes for disease development scoring, with 7 days being used in this research work and 3 days after inoculation with *B. sorokiniana* conidia being used in [56].

The most practically interesting result in the study of *T. aethiopicum* is the extremely high frequency (15%) of *Pst*-resistant landraces of this species. Notably, Vavilov described *T*. *abissinicum* (=*T. aethiopicum*) as being highly resistant to stripe rust [57], but in this study, 85% of the accessions were susceptible to this disease. Among 197 landraces and cultivars of durum wheat accessions from Ethiopia, 3 were found to be highly resistant to *Pst* [71], but it is difficult to compare these data with those obtained in the present work because of the different classifications of tetraploid wheats adopted in different gene centers. 

There are currently more than 80 genes for resistance to *P. striiformis* f. sp. *tritici* [72], but most of them do not protect wheat against modern *Pst* populations. Five genes for *Pst* resistance (*Yr 5*, *Yr10*, *Yr15*, *Yr24*, and *Yr26)* have been described as being effective in the Russian Federation [31,73], but virulence against *Yr24* has now been shown in the south of Russia [32]. In this study, only *Yr5* was effective both in the seedling and the flag leaf stages of plant ontogenesis, indicating an extremely narrow genetic diversity for effective resistance; the susceptibility of lines previously assessed as resistant in the northwestern region of the RF [31] could be explained by drastic changes in the pathogen population’s genetic structure (possibly due to spores’ migration from other regions) or the dependence of phenotypic virulence expression on environmental factors [74].

So, fifty-five samples of *T. aethiopicum*, classified as resistant to *Pst* at both stages of plant growth, can be recommended as a very valuable gene pool for breeding durum and bread wheats after detailed genetic studies. In addition to its resistance to *Pst*, *T. aethiopicum* is considered a valuable initial material for breeding to provide a purple grain color, lodging and drought tolerance, resistance to stem rust and *Cercosporella herpotrichoides* (Fron) Deighton. root rot, and a high protein content [75,76]. Grains of a purple color contain antioxidants that are beneficial for human health and are therefore of interest for their use in functional nutrition.

Spelt has a rich grain content of trace elements, vitamins, essential amino acids, and other useful substances. It is considered to be a very important crop in organic farming. It should have a leading role in obtaining ecologically clean grains [77]. Previously, 33 Ethiopian *T. dicoccum* samples were screened for resistance to *Bgt*, and ten were resistant to the disease [78], three (k-18971, k-18975, and k-19577) of which were confirmed to have this trait in this study; the susceptibility of the previously selected entries is likely related to differences in the populations of *Bgt* used for inoculation. Among the new entries, only k-67398 was found to be resistant. Two landraces (k-67397 and k-18971) of the species out of the forty-one under study were highly tolerant to Al^3+^. So, *T. dicoccum* sample k-18971 from Ethiopia was simultaneously resistant to *Bgt* and the abiotic stress factor. One landrace, k-67397, was resistant to *Pt* at 20 °C but showed a susceptible reaction at 25 °C. All entries were susceptible to HLB, including two previously described as resistant [56]. As a result, *T. dicoccum* samples resistant to *Bgt* and Al^3+^ can be used in the breeding of tetraploid and hexaploid wheats.

Among six Ethiopian landraces of *T. polonicum*, all were susceptible to the diseases, but one (k-43765) was highly resistant to aluminum ions.

So, we confirmed our previous conclusions on the absence of genetic diversity for effective resistance to *Pt*, SNB, and HLB among tetraploid and hexaploid wheat species [19,20,21] and among landraces of bread wheat [65] but identified samples with a high level of resistance to *Bgt*, *Pst*, and Al^3+^.

Ethiopian landraces of four wheat species possessing resistance to powdery mildew, stripe rust, and toxic aluminum ions could be recommended as initial materials for the breeding of resistant varieties. 

Since the samples resistant to these diseases can be protected by identical genes and the involvement of each of them in breeding will lead to the creation of a genetically homogeneous initial material, it is necessary to conduct genetic studies before their widespread use in the breeding process.

## 5. Conclusions

The genetic diversity of bread wheat for effective resistance to leaf rust, powdery mildew, Septoria nodorum blotch, dark-brown leaf spot blotch, and yellow rust is extremely narrow; the same situation exists for tolerance to aluminum abiotic stress. Broadening this diversity is a very urgent task for breeding purposes in order to reduce yield losses and to improve grain quality in modern wheat varieties. Samples of wheat landraces from Ethiopia and Eritrea from the VIR collection are of great importance in performing this task for several reasons. According to Vavilov, the genetic diversity for disease resistance is concentrated in gene centers of crops’ origin, with the region of Ethiopia and Eritrea being the secondary gene center of *Triticum*’s origin. The soils of the region are mainly acidic, and selection for aluminum toxicity tolerance could lead to the accumulation of resistant landraces. Bread wheat and its relatives of other origins from the collection have been extensively screened for the breeding of many valuable traits, including resistance to these diseases and Al^3+^ tolerance, while wheat samples from Ethiopia and Eritrea have never been studied.

The complete collection of four wheat species from these regions was screened for resistance to *Pt*, *Bgt*, SNB, and HLB. All samples of *T. aethiopicum* were evaluated for *Pst* resistance, and those of *T. aestivum*, *T. polonicum*, and *T. dicoccum* were evaluated for Al^3+^ tolerance. The findings reveal that bread wheat landraces from the region are susceptible to the studied diseases. In contrast, 20 entries of *T. aestivum* were highly tolerant to aluminum ions. As for *Bgt* resistance, four accessions of *T. dicoccum* were found to possess a high level of trait expression. A high proportion of *T. aethiopicum* entries (15%) were resistant to *Pst*. Samples with a tolerance to harmful aluminum ions were identified in *T. polonicum* and *T. dicoccum*, too.

The landraces from ETH and ERI, identified as being resistant to certain biotic and abiotic stresses, are of great interest for the breeding of different wheat species, especially considering the absence or extremely low genetic diversity for resistance in bread wheat of other origins and in durum wheat. We suggest that genetic studies of the selected samples should be performed before their widespread use in practical breeding.

All samples under study were susceptible to *Pt*, HLB, and SNB, indicating the necessity of searching for resistant samples in other parts of the VIR collection and possibly developing additional methods for controlling these diseases in wheat production.

## Figures and Tables

**Figure 1 plants-13-01166-f001:**
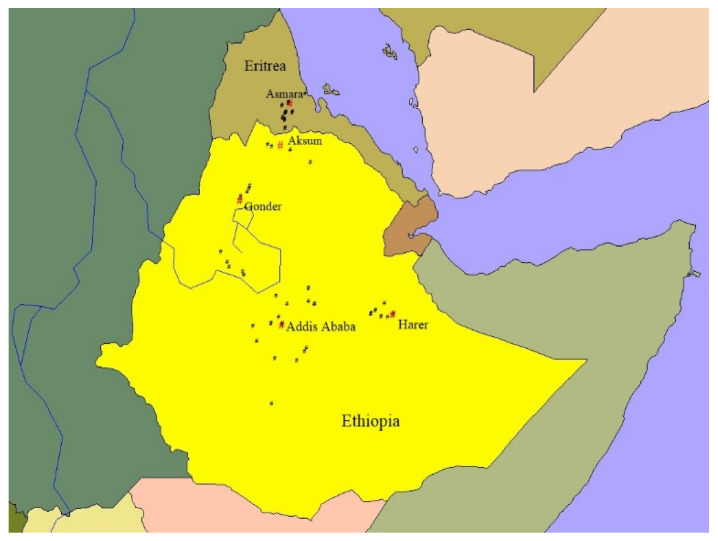
Sampling sites of spring bread wheat landraces in Ethiopia and Eritrea. Red “#”—city/town and black “#”—wheat sampling sites.

**Figure 2 plants-13-01166-f002:**
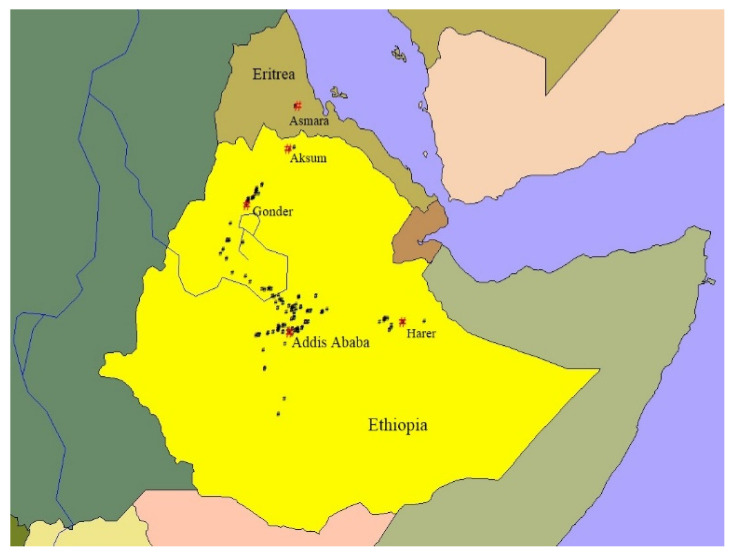
Sampling sites of *T. aethiopicum* landraces in Ethiopia and Eritrea. Red “#”—city/town and black “#”—wheat sampling sites.

**Figure 3 plants-13-01166-f003:**
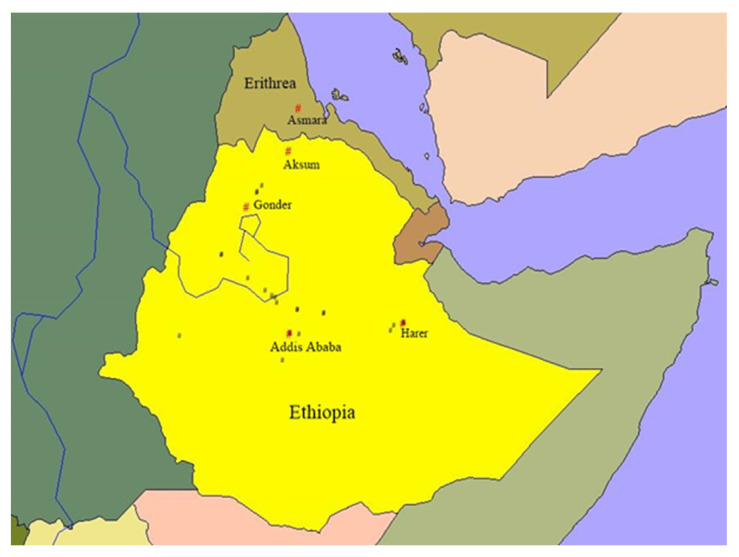
Sampling sites of *T. dicoccum* landraces in Ethiopia. Red “#”—city/town and black “#”—wheat sampling sites.

**Figure 4 plants-13-01166-f004:**
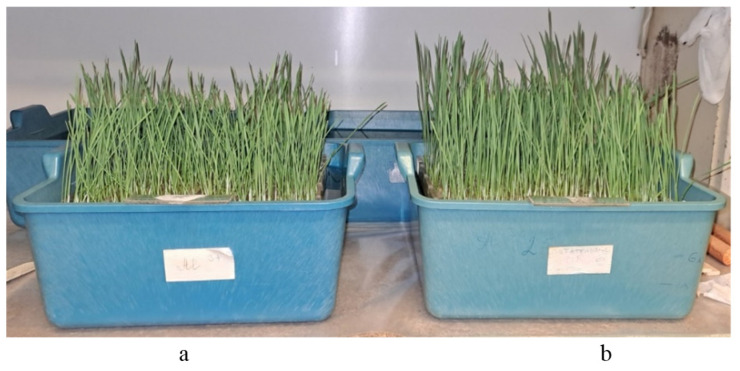
Growing of wheat seedlings in germinating boxes to evaluate aluminum ion resistance: (**a**) solution with AlCl_3_ × 6H_2_O; (**b**) water.

**Figure 5 plants-13-01166-f005:**
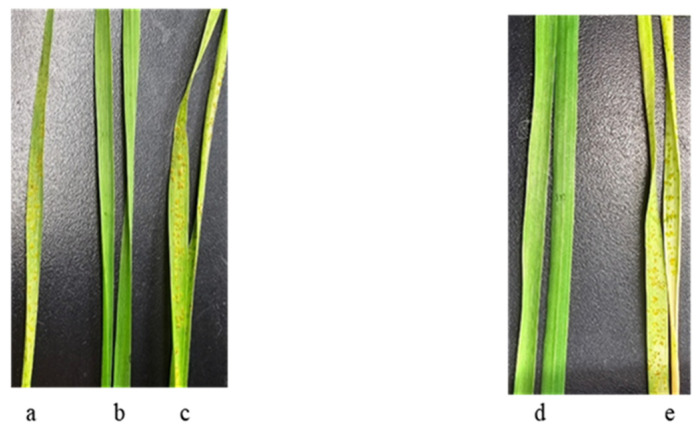
*Pt* development on wheat seedlings: (**a**) *T. aestivum* cv. Leningradka, 20 °C; (**b**) *T. aethiopicum* k-68236, 20 °C; (**c**) *T. aethiopicum* k-68236, 25 °C; (**d**) *T. dicoccum* k-67397, 20 °C; and (**e**) *T. dicoccum* k-67397, 25 °C.

**Figure 6 plants-13-01166-f006:**
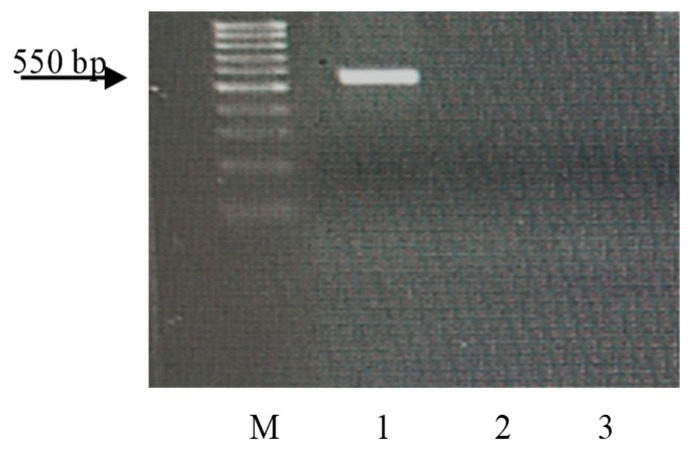
Amplicons of marker SCS5_550_ of the *Pt* resistance gene *Lr9*. M—100 bp DNA Ladder; 1—Th*Lr9*; 2—*T*. *dicoccum* k-67397; and 3—*T*. *aethiopicum* k-68236.

**Figure 7 plants-13-01166-f007:**
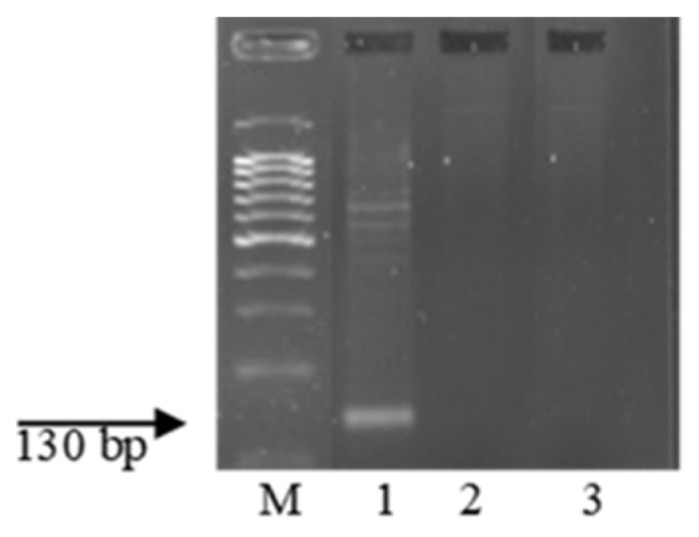
Amplicons of marker Gb of the *Pt* resistance gene *Lr19*. M—100 bp DNA Ladder; 1—Th*Lr19*; 2—*T. dicoccum* k-67397; and 3—*T. aethiopicum* k-68236.

**Figure 8 plants-13-01166-f008:**
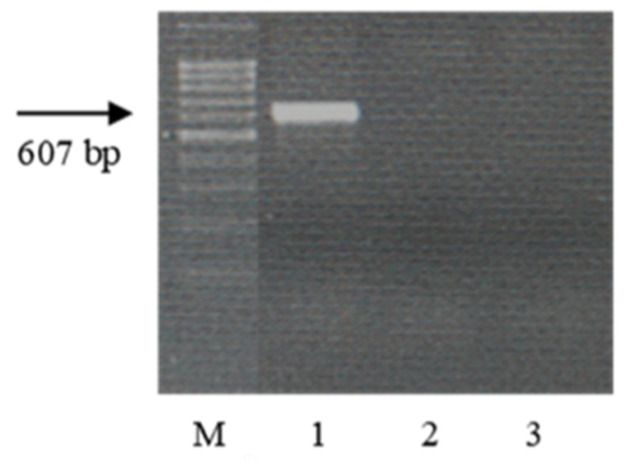
Amplicons of marker SCS1302_607_ of the *Pt* resistance gene *Lr24*. M—100 bp DNA Ladder; 1—Th*Lr24*; 2—*T. dicoccum* k-67397; and 3—*T. aethiopicum* k-68236.

**Figure 9 plants-13-01166-f009:**
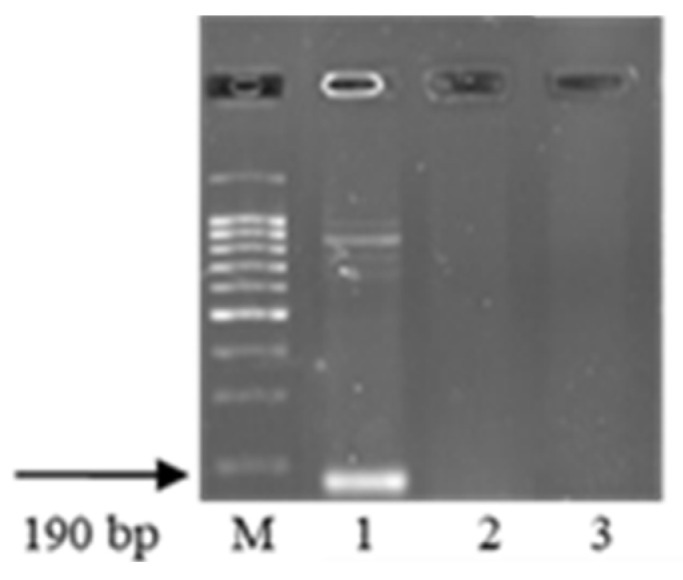
Amplicons of marker GDM35 of the *Pt* resistance gene *Lr41*. M—100 bp DNA Ladder; 1—KS90WGRC10; 2—*T. dicoccum* k-67397; and 3—*T. aethiopicum* k-68236.

**Figure 10 plants-13-01166-f010:**
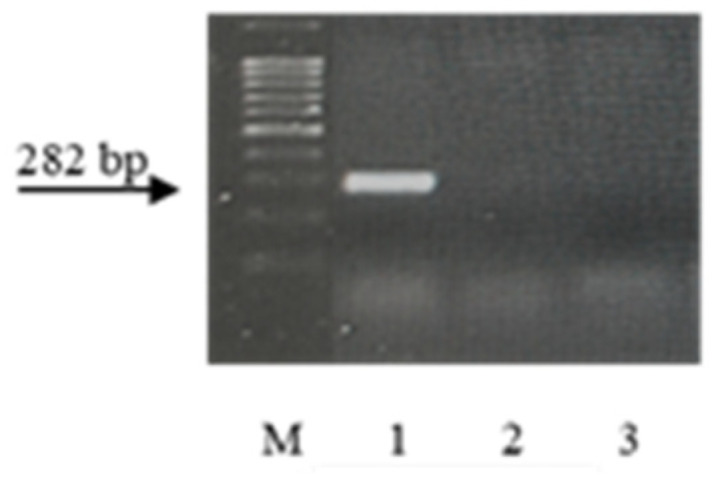
Amplicons of marker PS10 linked to the *Pt* resistance gene *Lr47*. M—100 bp DNA Ladder; 1—Pavon *Lr47*; 2—*T. dicoccum* k-67397; and 3—*T. aethiopicum* k-68236.

**Figure 11 plants-13-01166-f011:**
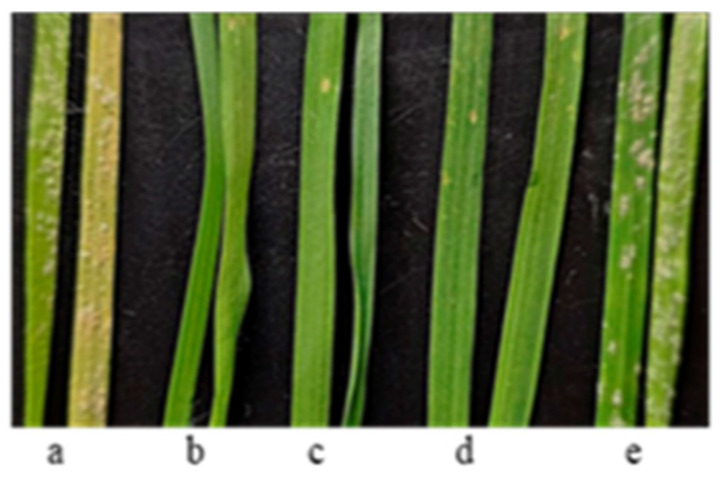
Powdery mildew development on the seedlings of *T. dicoccum* samples from ETH: (**a**) k-67397 (susceptible); (**b**) k-18971; (**c**) k-18975; (**d**) k-67398; and (**e**) k-13893 (susceptible).

**Figure 12 plants-13-01166-f012:**
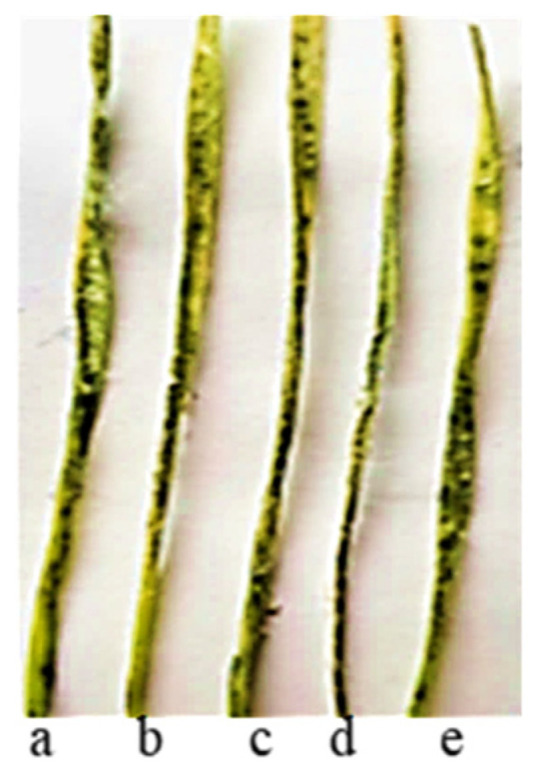
HLB’s development on seedlings of *T. dicoccum* and *T. aethiopicum* samples after infection with a mixture of *B. sorokiniana* isolates: (**a**) *T. dicoccum* k-13893; (**b**) *T. dicoccum* k-19596; (**c**) *T. dicoccum* k-19670; (**d**) *T. aethiopicum* k-19648; and (**e**) *T. aethiopicum* k-19287 (all previously identified as resistant [56]).

**Figure 13 plants-13-01166-f013:**
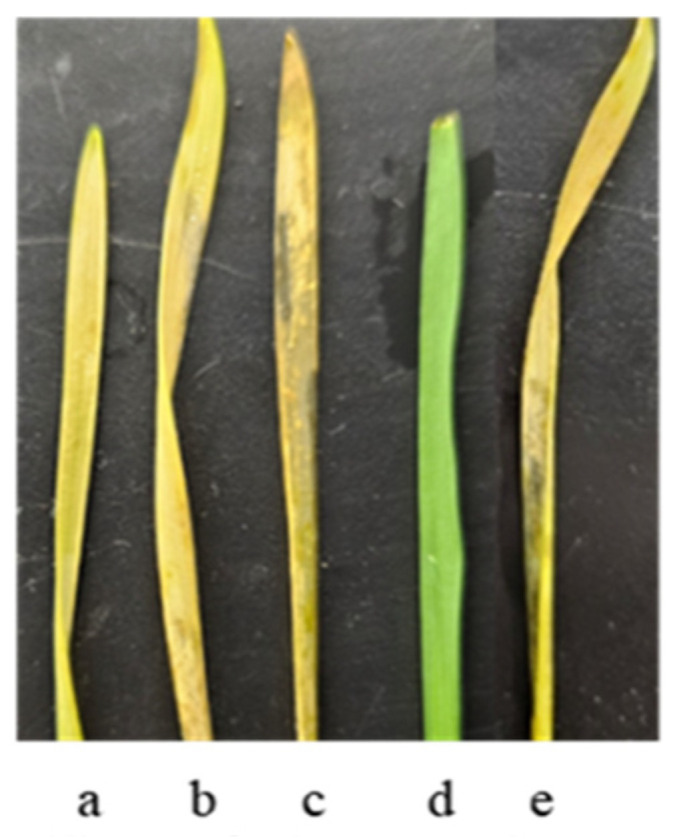
SNB’s development on seedlings of wheat accessions from Ethiopia and Eritrea: (**a**) *T. dicoccum* k-13893; (**b**) *T. dicoccum* k-19670; (**c**) *T. aethiopicum* k-19648; (**d**) *T. aestivum* k-43749—non infected; and (**e**) *T. aestivum* k-43749.

**Figure 14 plants-13-01166-f014:**
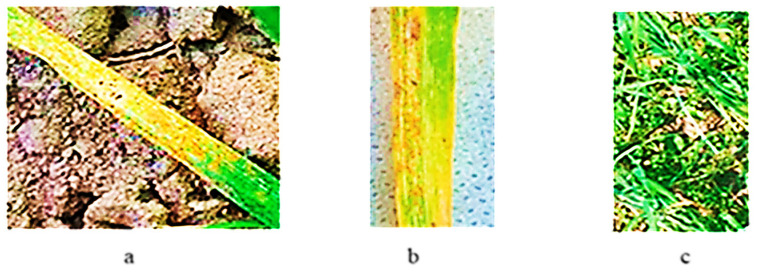
Seedlings of *T. aethiopicum* samples after infection with a population of *P. striiformis* f. sp. *tritici*: (**a**) k-68236; (**b**) k-68237; (**c**) k-68238.

**Figure 15 plants-13-01166-f015:**
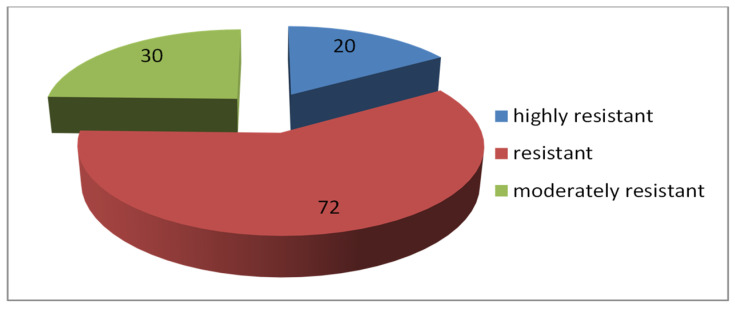
Numbers of bread wheat samples from Ethiopia and Eritrea with different level of aluminum resistance.

**Figure 16 plants-13-01166-f016:**
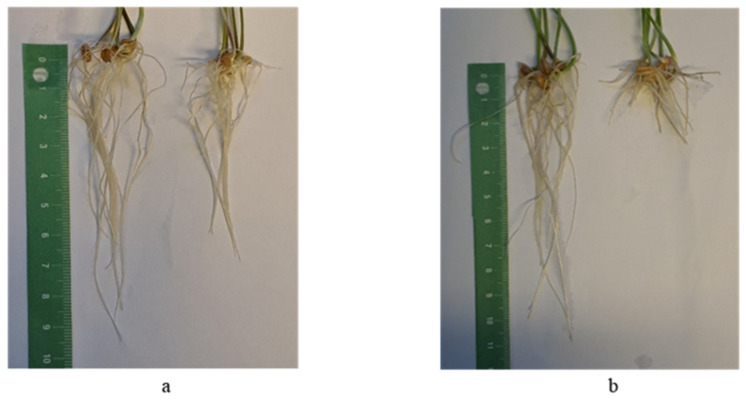
Seedling roots of two bread wheat samples differing in Al^3+^ resistance: (**a**) k-67822 (highly resistant, IRL = 0.98); (**b**) k-68127 (moderately resistant, IRL = 0.47). Left in each photo—control (water); right—solution with 185 µM Al^3+^.

**Figure 17 plants-13-01166-f017:**
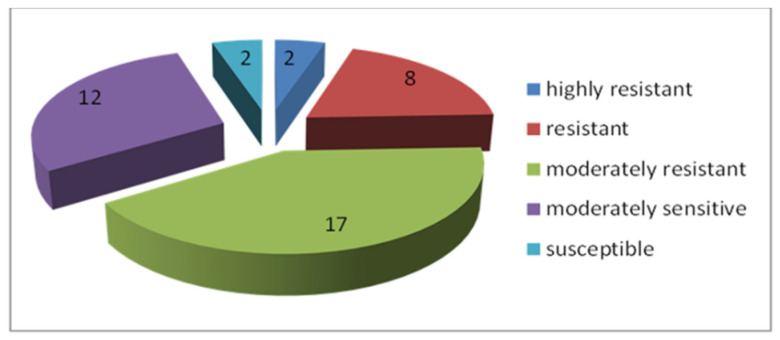
Number of *T. dicoccum* samples from Ethiopia and Eritrea, belonging to different groups of aluminum resistance according to IRL.

**Table 1 plants-13-01166-t001:** VIR’s wheat collections from Ethiopia and Eritrea were studied for resistance to diseases and acid tolerance.

Wheat Species	Number of Accessions		
	from Ethiopia	from Eritrea	Total
*T. aestivum* L.	62	60	122
*T. aethiopicum* Jakubz.	326	14	340
*T. dicoccum* (Schrank.) Schuebl.	40	1	41
*T. polonicum* L.	6	-	6
Total	434	75	509

**Table 3 plants-13-01166-t003:** Samples of *T. aethiopicum* highly resistant to yellow rust.

VIR Catalogue No. kk-	Origin	Collecting Site	Latitude	Longitude	Height, m	Types of Reaction to *Pst*	
						Seedlings	Flag Leaves
18983	ETH	t. Harer	E 42 07	N 09 18	-	0	0
18987	ETH	t. Harer	E 42 07	N 09 18	-	0	0
43776	ERI	t. Asmara, experimental station	E 38 55	N 15 20	2317	0	0
44521	ETH	Oromia, near from t. Addis-Abba	E 38 44	N 09 01	-	0	0
45990	ERI	t. Asmara, experimental station	E 38 55	N 15 20	2317	0	0
61012	ETH	Oromia, region t. Addis-Ababa, way to NW	E 38 33	N 09 13	2470	0	0
61021	ETH	Oromia, region t. Addis-Ababa	E 38 33	N 09 03	2580	0	0
61025	ETH	Oromia, region t. Addis-Ababa, way to NW	E 38 36	N 09 12	2440	0	0
61028	ETH	Oromia, region t. Addis-Ababa, way to NE	E 39 14	N 09 18	2760	0	0
61048	ETH	Oromia, region t. Adis-Alem	E 38 26	N 09 09	2700	0	0
61049	ETH	Oromia, region t. Adis-Alem	E 38 26	N 09 09	2700	0	0
61053	ETH	Oromia, region t. Addis-Ababa, way to NW	E 38 33	N 09 12	2540	0	0
61396	ETH	Oromia, region t. Addis-Ababa, way to NE	E 39 14	N 09 18	2760	0	0
61409	ETH	Oromia, region t. Addis-Ababa, way to N	E 38 50	N 09 42	2560	0	0
61411	ETH	Oromia, region t. Addis-Ababa, way to N	E 38 50	N 09 42	2560	0	0
61416	ETH	Oromia, region t. Addis-Ababa	E 38 33	N 09 03	2580	0	0
61422	ETH	Region Shewa	E 38 00	N 08 00	2180	0	0
61430	ETH	Region Shewa	E 38 00	N 08 00	2180	0	0
65969	ETH	Oromia, along the road Arusi	E 39 45	N 07 57	2720	0	0
67875	ETH	Amhara, district Amba Gorgis	E 37 41	N 12 48	2750	0	0
67876	ETH	Amhara, Goja, district Dangola	E 36 46	N 11 21	2160	0	0
67879	ETH	Harari, district Celenco haro	E 41 36	N 09 24	2244	0	0
67883	ETH	Oromia, district Segen, village Ana Degum	E 38 33	N 09 51	2913	0	0
67885	ETH	Harari, village Chelenko	E 41 34	N 09 21	2155	0	0
67891	ETH	Amhara, 9 km north of Addis-Ababa	E 38 42	N 09 07	2530	0	0
67892	ETH	Amhara, a market near the city Addis-Ababa	E 38 50	N 09 07	2639	0	0
67897	ETH	Amhara, a market near the city Addis-Ababa	E 38 50	N 09 07	2639	0	0
67898	ETH	Amhara, a market near the city Addis-Ababa	E 38 50	N 09 07	2639	0	0
67901	ETH	Amhara, Gondor, district Debark, village Deber	E 37 56	N 13 08	2830	0	0
68208	ETH	Amhara, Gondor, district Gondor Zura	E 37 30	N 12 44	2806	0	0
68212	ETH	Amhara, Gondor, district Amba Gorgis	E 37 41	N 12 48	2697	0	0
68217	ETH	Oromia, the road to Debre Marcos-Features	E 38 14	N 10 02	2276	0	0
68222	ETH	Amhara, on the way Addis-Ababa-Debre-Birhan	E 39 17	N 09 19	2876	0	0
68223	ETH	Amhara, on the way Addis-Ababa-Debre-Birhan	E 39 17	N 09 19	2876	0	0
68228	ETH	Oromia, district Chencho	E 38 55	N 09 05	2442	0	0
68230	ETH	Harari, district Hula Jeneta	E 41 48	N 09 09	2450	0	0
68231	ETH	Harari, district Chelenko	E 41 36	N 09 24	-	0	0
68233	ETH	Amhara, near the city Bahir Dar	E 36 54	N 11 35	2048	0	0
68238	ETH	Amhara, Gondor, district Tikil Dingay	E 37 30	N 12 43	2675	0	0
68241	ETH	Amhara, Gondor, district Gondor Zura	E 37 30	N 12 44	2806	0	0
68244	ETH	Amhara, (Gondor), district Amba Giorgis	E 37 41	N 12 48	2750	0	0
68249	ETH	Amhara, district Ancober, village Ancober Cheffa	E 39 44	N 09 35	2720	0	0
68264	ETH	Amhara, district Segen, village Yetenora	E 38 08	N 10 16	2330	0	0
68267	ETH	Amhara, district Ejere	E 38 24	N 10 04	2200	0	0
68269	ETH	Oromia, district Segen, village Ana Degum	E 38 33	N 09 51	2913	0	0
68270	ETH	Oromia, district Gerar Jarso, village Chage/Worku	E 38 48	N 09 39	2496	0	0
68273	ETH	Oromia, district Debre Tsige, village Tere	E 38 48	N 09 39	2624	0	0
68281	ETH	Oromia, district Chencho, v. Gende Gorfo	E 38 52	N 09 26	2597	0	0
68283	ETH	Oromia, district Chencho, v. Gende Gorfo	E 38 52	N 09 26	2597	0	0
68284	ETH	Oromia, district Chencho	E 38 55	N 09 05	2442	0	0
68285	ETH	Oromia, district Chencho	E 38 55	N 09 05	2442	0	0
68290	ETH	Amhara, district Ancober	E 39 44	N 09 35	2622	0	0

**Table 4 plants-13-01166-t004:** Accessions of bread wheat from Ethiopia and Eritrea with a high seedling tolerance to Al^+3^ toxic ions.

VIR Catalogue No. kk-	Origin	Collecting Site	Latitude	Longitude	Height, m	Index of Root Length
43749	ETH	Oromia, t. Yirga-Alem, market	E 38 24	N 06 44	1755	0.87
43755	ETH	Oromia, near from t. Addis-Alem	E 32 56	N 09 02	2274	0.85
43760	ETH	Amhara, t. Gonder, market	E 37 27	N 12 36	2135	0.83
43775	ERI	Maekel, Experimental station, t. Asmara	E 38 55	N 15 20	2317	0.85
43777	ERI	Maekel, Experimental station, t. Asmara	E 38 55	N 15 20	2317	0.84
44512	ETH	Oromia, t. Addis-Alem, 30 km to E from t. Ambo	E 38 23	N 09 02	-	0.81
65893	ETH	Tigre, 20 km to NW from t. Mek’le	E 39 35	N 13 38	2260	0.86
65894	ETH	Tigre, v. Marafluba	E 38 58	N 14 00	1800	0.95
67822	ETH	Amhara, region Yalika	E 37 29	N 12 40	2031	0.98
67831	ETH	Amhara, region Debre Berkhan, 2 km to East	E 39 33	N 09 40	-	0.83
67992	ERI	Asmara, v. Hezega	E 38 57	N 15 20	2317	0.83
67997	ERI	v. Mendefera	E 38 48	N 14 52	1909	0.84
67998	ERI	v. Dubaruwa	E 38 51	N 15 04	1856	0.81
67999	ERI	v. Mendefera	E 38 48	N 14 52	1933	0.84
68104	ERI	Asmara, v. Hezega	E 38 57	N 15 20	2062	0.81
68105	ERI	v. Mendefera	E 38 48	N 14 52	1898	0.85
68108	ERI	t. Dek’emhare	E 39 03	N 15 04	1904	0.88
68109	ERI	Asmara, v. Hezega	E 38 57	N 15 20	2176	0.82
68113	ERI	–	–	–	–	0.91
68118	ERI	Asmara, v. Hezega	E 38 57	N 15 20	2337	0.95

## Data Availability

Data are contained within the article.

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
