# Peer review of "Genetic Diversity for Effective Resistance in Wheat Landraces from Ethiopia and Eritrea to Fungal Diseases and Toxic Aluminum Ions"

_plants, 2024, doi:10.3390/plants13081166_

Round 1
Reviewer 1 Report
Comments and Suggestions for Authors
- Authors could cite the GIS tool were used to draw the maps in figures 1, 2 and 3
- Authors could specify in table 2 the assignment of each marker to chromosomes
- If possible, authors could replace figures 12 and 14 with other figures with higher resolution
Author Response
Answer to reviewer 1
- Authors could cite the GIS tool were used to draw the maps in figures 1, 2 and 3.
We added this information.
- Authors could specify in table 2 the assignment of each marker to chromosomes.
We added this information.
- If possible, authors could replace figures 12 and 14 with other figures with higher resolution.
We replaced these figures.
We would like to thank the reviewer for the remarks.
Reviewer 2 Report
Comments and Suggestions for Authors
The article, titled "Exploring Genetic Diversity for Enhanced Resistance in Wheat Landraces from Ethiopia and Eritrea to Combat Fungal Diseases and Aluminum Toxic Ions," delves into the genetic diversity of wheat landraces from these regions. The primary focus is on understanding their mechanisms of resistance against fungal diseases and aluminum toxicity. This study significantly contributes to the crucial field of crop improvement, especially in regions where wheat is a staple food.
In general, the paper significantly contributes to advancing and comprehension of genetic diversity in wheat landraces and their mechanisms of resistance. Implementing the recommended enhancements has the potential to further amplify the paper's influence and relevance within the realms of crop science and agriculture.
Introduction section is well written and properly addressed the problems. The citation used in the section is updated.
Materials & methods sections:
The materials and methods section also well written.
Results:
The research findings are aptly conveyed in the paper through well-structured tables, figures, and succinct textual descriptions, facilitating a clear and easily understandable interpretation of the results. However, there is need to change the text format in the figure.
There is a need to add the future concerns segment incorporating on prospective research directions would provide guidance to researchers and policymakers seeking to expand upon the existing findings. Offering insights into potential areas for further investigation would enhance the paper's overall value.
Author Response
Answer to reviewer 2
- The research findings are aptly conveyed in the paper through well-structured tables, figures, and succinct textual descriptions, facilitating a clear and easily understandable interpretation of the results. However, there is need to change the text format in the figure.
We did it.
- There is a need to add the future concerns segment incorporating on prospective research directions would provide guidance to researchers and policymakers seeking to expand upon the existing findings. Offering insights into potential areas for further investigation would enhance the paper's overall value.
We added this information in the article text.
We thank deeply the reviewer for favorable impression of the work.
Reviewer 3 Report
Comments and Suggestions for Authors
The study aims to unveil the genetic diversity for effective resistance to various foliar diseases and aluminum toxic ions in wheat landraces from Ethiopia and Eritrea. It analyzes 509 samples of four wheat species using a combination of field and laboratory experiments, including inoculations with fungal pathogens and exposure to aluminum ions. The findings highlight specific entries showing resistance to certain pathogens and aluminum toxicity.
-
English Language Quality and Clarity: The manuscript requires substantial improvements in language quality and clarity. Several sentences are convoluted and lack coherence, making comprehension difficult. I suggest a thorough proofreading and restructuring of sentences for better readability and clarity.
-
Literature Review and Novelty: The literature review is outdated and lacks references to recent studies in the field. To enhance the manuscript's novelty and relevance, the authors should cite recent research articles that address genetic diversity, disease resistance mechanisms, and aluminum toxicity tolerance in wheat. I recommend including references to the following papers:
- https://www.sciencedirect.com/science/article/pii/S2405844023068512
- https://www.sciencedirect.com/science/article/pii/S101836472300040X
- https://www.nature.com/articles/s41598-021-02594-4
-
Research Gaps and Justification: The manuscript should clearly outline the research gaps addressed by the study and justify its significance in the context of existing literature. The authors should discuss how their findings contribute to filling these gaps and advancing knowledge in wheat breeding and genetics.
-
Discussion Section: The discussion section requires expansion and integration of recent literature to contextualize the study's findings. The authors should compare and contrast their results with previous research, discuss the implications of resistant entries identified, and propose future research directions.
-
Conclusion: The manuscript lacks a concrete conclusion section. A well-defined conclusion summarizing the key findings and their implications is essential. Additionally, the authors should provide insights into potential applications of the resistant entries identified and outline future prospects for wheat breeding programs.
Recommendations for Revision:
- Conduct thorough proofreading to enhance language quality and clarity.
- Update the literature review with recent references to establish the novelty and relevance of the study.
- Clearly articulate research gaps and justify the study's significance.
- Expand the discussion section by integrating recent literature and discussing the implications of the findings.
- Add a comprehensive conclusion section summarizing key findings and outlining future prospects for research and wheat breeding programs.
Overall, the manuscript has the potential to contribute significantly to the understanding of genetic diversity and disease resistance in wheat landraces. However, substantial revisions are necessary to improve its clarity, relevance, and scholarly impact.
Comments on the Quality of English LanguageIt needs improvements
Author Response
-
Answer to reviewer 3
- English Language Quality and Clarity: The manuscript requires substantial improvements in language quality and clarity. Several sentences are convoluted and lack coherence, making comprehension difficult. I suggest a thorough proofreading and restructuring of sentences for better readability and clarity.
We sent the article to 2 native English speakers and made all correction according to their remarks.
- Literature Review and Novelty: The literature review is outdated and lacks references to recent studies in the field. To enhance the manuscript's novelty and relevance, the authors should cite recent research articles that address genetic diversity, disease resistance mechanisms, and aluminum toxicity tolerance in wheat. I recommend including references to the following papers:
- https://www.sciencedirect.com/science/article/pii/S2405844023068512
- https://www.sciencedirect.com/science/article/pii/S101836472300040X
- https://www.nature.com/articles/s41598-021-02594-4
Unfortunately we did not find any relation of these articles to the topic of our article. Some references we added in the article
- Research Gaps and Justification: The manuscript should clearly outline the research gaps addressed by the study and justify its significance in the context of existing literature. The authors should discuss how their findings contribute to filling these gaps and advancing knowledge in wheat breeding and genetics.
We tried to add this information.
- Discussion Section: The discussion section requires expansion and integration of recent literature to contextualize the study's findings. The authors should compare and contrast their results with previous research, discuss the implications of resistant entries identified, and propose future research directions.
We tried to add this information.
- Conclusion: The manuscript lacks a concrete conclusion section. A well-defined conclusion summarizing the key findings and their implications is essential. Additionally, the authors should provide insights into potential applications of the resistant entries identified and outline future prospects for wheat breeding programs.
We added the text of the conclusion.
Recommendations for Revision:
- Conduct thorough proofreading to enhance language quality and clarity.
We did it
- Update the literature review with recent references to establish the novelty and relevance of the study.
We insist that recent references related to the work were used.
- Clearly articulate research gaps and justify the study's significance.
We did it.
- Expand the discussion section by integrating recent literature and discussing the implications of the findings.
We added into discussion the significance of identified resistant wheat for breeding.
- Add a comprehensive conclusion section summarizing key findings and outlining future prospects for research and wheat breeding programs.
We write conclusion.
Overall, the manuscript has the potential to contribute significantly to the understanding of genetic diversity and disease resistance in wheat landraces. However, substantial revisions are necessary to improve its clarity, relevance, and scholarly impact.
We tried to do it.
Comments on the Quality of English Language
It needs improvements.
We checked it.
We would like to thank the reviewers for thoroughfull reading and editing of the article.
Reviewer 4 Report
Comments and Suggestions for Authors
Please see the attached file for comments.

Author Response
Answer to reviewer 4
- The research materials of wheat are relatively few, not representative, and the samplings are relatively random, so it is difficult to find the characteristics of variety resistance for the mentioned-aboved stresses.
The material from our viewpoint is quite extensive – 509 samples. It is the complete VIR collection of wheats from ETH and ERI. It is not random because represents all regions of wheat growing in these countries. May by it is difficult but we gave these characteristics.
- The author selected 5 wheat diseases for screening its resistance ,respectively. This five kinds of disease resistance genetic characteristics are very different, and the mechanisms are also very complex, plus the screening for wheat resistance of aluminum toxicity. Although the data in this manuscript are very big, the conclusion at present also failed to see from the existing data both for so many biotic and abiotic stress. The number of analysis for these stresses is lack of biological significence. Furthoremore, some research has not finished. Therefore, which has the little orlimited use for the readers.
The significance of the results does not depend on resistance mechanisms. We identified new plant materials with high level of resistance that are very useful for breeding. Conclusion was written. We outlined the significance of the work to be continued. We cannot agree that the article has the little or limited use for the readers: wheat breeders and geneticists should be interested in the information.
- The depth of research in this manuscript is also low. There are also the lack of close links between the morphological indicators and the molecular markers of existing genes related to disease resistance, and no new linked molecular markers developed. Therefore, it is also hard for molecular breeding to use in the future. The practise significence is also limited.
The task of the work was to give characteristics for resistance and to identify new resistant genotypes. The task was performed. To develop molecular markers is absolutely different task.
Reviewer 5 Report
Comments and Suggestions for Authors
The authors aim to "The purpose of this research was to study resistance to five diseases and acid tolerance in Triticum L. species from the VIR collection originated from Ethiopia and Eritrea and to identify entries with high level of the trait’s expression." Yet, it is very hard to understand which diseases were studied or how acid tolerance was followed. There is also no clear explanation of how many replicates per line were used.
The discussion is very poorly written and overall, a duplication of results.
Comments on the Quality of English LanguageThe manuscript needs to be text edited since there are multiple mistakes and grammatical typos.
Author Response
Answer to reviewer 5
1.The authors aim to "The purpose of this research was to study resistance to five diseases and acid tolerance in Triticum L. species from the VIR collection originated from Ethiopia and Eritrea and to identify entries with high level of the trait’s expression." Yet, it is very hard to understand which diseases were studied or how acid tolerance was followed. There is also no clear explanation of how many replicates per line were used.
We studied five diseases, and they are listed in MM, results and discussions. Replication also were described (not less than 4).
- The discussion is very poorly written and overall, a duplication of results.
We rechecked the discussion and added some remarks.
- Comments on the Quality of English Language
The manuscript needs to be text edited since there are multiple mistakes and grammatical typos. We have done it.
Round 2
Reviewer 3 Report
Comments and Suggestions for Authors
Paper is improved and can be processed for acceptance
Comments on the Quality of English LanguageIts fine
Author Response
Answer to Reviewer 3
Minor editing of English language required.
We again sent the revised article to 3 new English native speakers and took into account all their comments.
|
Yes |
Can be improved |
Must be improved |
Not applicable |
|
|
Does the introduction provide sufficient background and include all relevant references? |
( ) |
(x) |
( ) |
( ) |
|
Are all the cited references relevant to the research? |
( ) |
(x) |
( ) |
( ) |
|
Is the research design appropriate? |
( ) |
(x) |
( ) |
( ) |
|
Are the methods adequately described? |
( ) |
(x) |
( ) |
( ) |
|
Are the results clearly presented? |
( ) |
(x) |
( ) |
( ) |
|
Are the conclusions supported by the results? |
( ) |
(x) |
( ) |
( ) |
We made some changes in the text.
I would like to thank the reviewer for thorough reading of the text and very valuable remarks.
Reviewer 5 Report
Comments and Suggestions for Authors
This second version has not changed significantly when compared with the first one. I have previously made several comments/doubts, for which authors have a made a little attempt to answer, or to change the manuscript. Those were the following:
"1.The authors aim to "The purpose of this research was to study resistance to five diseases and acid tolerance in Triticum L. species from the VIR collection originated from Ethiopia and Eritrea and to identify entries with high level of the trait’s expression." Yet, it is very hard to understand which diseases were studied or how acid tolerance was followed. There is also no clear explanation of how many replicates per line were used."
The authors answered: "We studied five diseases, and they are listed in MM, results and discussions. Replication also were described (not less than 4)." However, this is actually not comprehensive in the text. In fact, the authors have now added: "Each sample was studied in three replications." - This not only contradicts the authors answer but also reveals a very low number of replicates.
" 2. The discussion is very poorly written and overall, a duplication of results.".
The authors answered: "We rechecked the discussion and added some remarks." However, my comment is still the same. There is now generalization of results or discussion concerning the resistance found in some genotypes, but not others, or the reason for the traits involved. In addition, the authors have now added more references, and the manuscript has an excessive amount of citations (80) - including a very high amount of self-citations.
I also stated that " Comments on the Quality of English Language: The manuscript needs to be text edited since there are multiple mistakes and grammatical typos."
The authors replied: "We have done it." This is simply not true.
Overall, the manuscript lacks clarity and specific aims of why the study was performed, and specifically how it was performed.
Comments on the Quality of English LanguageSee above.
Author Response
Answer to Reviewer 5
|
Does the introduction provide sufficient background and include all relevant references? |
( ) |
( ) |
(x) |
( ) |
|
Are all the cited references relevant to the research? |
( ) |
( ) |
(x) |
( ) |
|
Is the research design appropriate? |
( ) |
( ) |
(x) |
( ) |
|
Are the methods adequately described? |
( ) |
( ) |
(x) |
( ) |
|
Are the results clearly presented? |
( ) |
( ) |
(x) |
( ) |
|
Are the conclusions supported by the results? |
( ) |
( ) |
( ) |
( ) |
Introduction describes the problem of genetic diversity in wheat for resistance to biotic and abiotic stresses, the necessity of its broadening and why it is necessary to study the proposed plant material. Its origin in the collection is also presented.
References were chosen concerning all the work aspects.
The research design was chosen to aim the general purpose – to identify wheat samples with high level of resistance to harmful diseases and aluminum ions.
We described all methods we used. In some cases, we refer to our previous works to reduce the repetition rate (request of editor).
The result is identification of resistant samples, and we gave full information on their origin. In the case of genes for leaf rust resistance results on their molecular identification were also presented.
We suppose that is convenient for the article describing genetic diversity of important breeding traits. From our opinion it is confirmed by positive reviews from other 4 reviewers
Comments and Suggestions for Authors
This second version has not changed significantly when compared with the first one. I have previously made several comments/doubts, for which authors have a made a little attempt to answer, or to change the manuscript. Those were the following:
"1.The authors aim to "The purpose of this research was to study resistance to five diseases and acid tolerance in Triticum L. species from the VIR collection originated from Ethiopia and Eritrea and to identify entries with high level of the trait’s expression." Yet, it is very hard to understand which diseases were studied or how acid tolerance was followed. There is also no clear explanation of how many replicates per line were used."
The authors answered: "We studied five diseases, and they are listed in MM, results and discussions. Replication also were described (not less than 4)." However, this is actually not comprehensive in the text.
Lines 35-40 – a list of diseases resistance to which we studied.
Line 123 – abbreviation of diseases resistance to which we studied. Here we added full names of the diseases.
Replication also were described (not less than 4). In fact, the authors have now added: "Each sample was studied in three replications." - This not only contradicts the authors answer but also reveals a very low number of replicates.
Line 149 – 15-25 plants in each experiment.
Line 155 – accession identified………in at least 3 independent experiments. We add “additional” 1+3=4
In fact, the authors have now added: "Each sample was studied in three replications."
Really, we added this sentence but concerning tolerance to aluminum.
" 2. The discussion is very poorly written and overall, a duplication of results.
As I think discussion is really discussion of obtained results and not abstract reviewing of problem not concerning the aim and ideas of the work
The authors answered: "We rechecked the discussion and added some remarks." However, my comment is still the same. There is now generalization of results or discussion concerning the resistance found in some genotypes, but not others, or the reason for the traits involved.
We wrote about the reasons to study the traits involved – resistance to harmful abiotic and biotic stresses, causing very big losses in wheat yield.
We discussed our results in view of modern concepts of wheat resistance and practical use of identified samples.
In addition, the authors have now added more references, and the manuscript has an excessive amount of citations (80) - including a very high amount of self-citations.
Some reviewers asked us to add references. We did it. This reviewer indicted to extra citation. We deleted 3 references to articles of Tyryshkin.
I also stated that " Comments on the Quality of English Language: The manuscript needs to be text edited since there are multiple mistakes and grammatical typos."
The authors replied: "We have done it." This is simply not true.
We sent the old version of the article to 3 English native speakers and changed all their remarks. Now we again sent 3 new English native speakers and took into account all their comments.
Overall, the manuscript lacks clarity and specific aims of why the study was performed, and specifically how it was performed.
We explained that genetic diversity of previously studied material for the diseases and aluminum was very low. We explained why we chose plant material that we studied. We described in detail all experimental methods. Then the editor wrote the repetition rate is very high, so we had to refer to earlier published articles.
Comments on the Quality of English Language
See above.
See above.